# Modification of Poly(Ethylene 2,5-Furandicarboxylate) with Poly(Ethylene glycol) for Biodegradable Copolyesters with Good Mechanical Properties and Spinnability

**DOI:** 10.3390/polym11122105

**Published:** 2019-12-14

**Authors:** Peng Ji, Danping Lu, Shengming Zhang, Wanying Zhang, Chaosheng Wang, Huaping Wang

**Affiliations:** 1Co-innovation center for textile industry, Shanghai 201620, China; jipeng@dhu.edu.cn; 2State Key Laboratory for Modification of Chemical Fibers and Polymer Materials, Key Laboratory of Textile Science & Technology (Ministry of Education), College of Materials Science and Engineering, Donghua University, Shanghai 201620, China; ludanpingya@163.com (D.L.); SMZhang_dhu@163.com (S.Z.); zwy15221833265@163.com (W.Z.)

**Keywords:** 2,5-furandicarboxylic acid, poly(ethylene glycol), hydrophilic, degradability, spinning

## Abstract

Using 2,5-furandicarboxylic acid, ethylene glycol, and poly(ethylene glycol) as raw materials and ethylene glycol antimony as a catalyst, poly(ethylene furandicarboxylate) (PEF) and polyethylene glycol (PEG) copolymers (PEGFs) were synthesized by transesterification by changing the molecular weight of PEG (from 600 to 10,000 g/mol) and the PEG content (from 10 to 60 wt %). The thermal, hydrophilic, degradation, and spinnility characteristics of these copolymers were then investigated. Thermogravimetric analysis shows that PEGF is thermally stable at 62 °C, much lower than the temperature for PEF. The intrinsic viscosity of the obtained copolyester was between 0.67 and 0.99 dL/g, which is higher than the viscosity value of PEF. The contact angle experiment shows that the hydrophilicity of PEGFs is improved (the surface contact angle is reduced from 91.9 to 63.3°), which gives PEGFs a certain degradability, and the maximum mass loss can reach approximately 15%. Melt spinning experiments show that the PEGF polymer has poor spinnability, but the mechanical properties of the polymer monofilament are better.

## 1. Introduction

Plastics synthesized from petroleum are common in human life, and white pollution has attracted increasing attention. The non-renewability of oil and its ultimate depletion and rising prices introduce a considerable challenge to the petroleum industry relying on petroleum resources. People have begun to focus on green polymer materials, and industry and academia have also increased efforts on the research and development of polymer materials based on renewable resources [1,2,3,4,5]. 

At present, widely used bio-based polymer materials mainly include polylactic acid (PLA) [6], poly(hydroxy fatty acid) (PHA) [7], poly(glycolic acid) (PGA) [8], and poly(butylene glycol succinate) (PBS) [9]. They are all aliphatic polymers. Due to the lack of a rigid aromatic ring structure in the molecular structure, their mechanical properties (such as strength and modulus) and heat resistance (such as thermomechanical properties and heat distortion temperature) are significantly lower than those of petroleum-based polymers such as poly(ethylene terephthalate) (PET), poly(carbonate) (PC), and aromatic nylon (PA), which severely restricts the range of applications of aliphatic polymers. Therefore, to partially replace petroleum-based polymer materials, there is an urgent need to introduce a rigid ring structure into the molecular structures of bio-based polymer materials to increase their strength through synthetic techniques. Poly(glycolic acid) is a kind of synthetic polymer material with good biodegradability and biocompatibility. Poly(glycolic acid) (PGA) is gradually degraded after being used for a certain period of time and is eventually converted to water and carbon dioxide which are harmless to the human body, animals, plants, and the natural environment. 2,5-furandicarboxylic acid (FDCA) is a diacid based on renewable resources whose structure is similar to that of terephthalic acid. FDCA is considered by the US Department of Energy to be one of the 12 important chemicals in the green chemical industry [10]. 2,5-furandicarboxylic acid is a member of the furan family and is available from renewable sources [11,12,13]. FDCA is abundant and can be obtained from fructose and galactose. FDCA has a structure similar to that of terephthalic acid and is an aromatic compound with a cyclic conjugated system containing two carboxyl groups [14,15]. Thus, FDCA is considered a substitute for terephthalic acid [16,17,18]. Related polyesters, such as poly(ethylene furandicarboxylate) (PEF) [13,16], poly(propylene furandicarboxylate) (PPF) [19,20,21], and poly(butanediol furandicarboxylate) (PBF) [22,23,24], have been extensively studied. As discussed above, the introduction of degradable monomers could be a feasible way to balance the degradability and other properties. Hu et al. reported the copolyesters of FDCA, lactic acid, and 1,4-butanediol, and their degradation behavior in phosphate buffered solution was evaluated and discussed [25].

In this work, a series of poly(ethylene 2,5-furandicarboxylate)-poly(ethylene glycol) (PEGFs) were synthesized from available biobased ethylene glycol (EG), 2,5-furandicarboxylic acid (FDCA), and poly(ethylene glycol) (PEG) through a two-step melt polycondensation method. The effect of the PEG content and different molecular weights on the crystallization behavior and hydrophilic, thermal, and mechanical properties were analyzed. In addition, the degradation behavior of PEGFs was determined by analyzing the weight loss and inherent viscosity with degradation time when exposed to a phosphate buffered solution.

## 2. Experimental

### 2.1. Materials

Dimethyl 2,5-furandicarboxylate(DMFD) (99.2%) was purchased from Ningbo Jisu New Material Technology Co., Ltd. (Ningbo, China). Ethylene glycol (CP grade) and chloroform (CP grade) were purchased from Shanghai Lingfeng Chemical Reagent Co., Ltd. (Shanghai, China). Polyethylene glycol (PEG) (*M*n = 600, 2000, 6000, 10,000 g/mol) was purchased from Sinopharm Pharmaceutical Co., Ltd. (Shanghai, China). Ethylene glycol antimony (Sb_2_(EG)_3_) (99%) was obtained from Hunan Chenzhou Mining Co., Ltd. (Hunan, China). Zinc acetate (99.5%), trimethyl phosphate (99%), trifluoroacetate (99%), and deuterated trifluoroacetic acid (TFA) were all purchased from Shanghai Titan Technology Co., Ltd. (Shanghai, China). Phenol (CP grade) and the tetrachloroethane (CP grade) were purchased from Sinopharm Chemical Reagent Co., Ltd. (Shanghai, China). Antioxidants 1010 were obtained from BASF. Co. (Shanghai, China). All the chemicals were used as received without further purification.

### 2.2. Synthesis of PEF and PEGFs

PEGFs were synthesized from ethylene glycol (EG), dimethyl 2,5-furandicarboxylate (DMFD), and poly(ethylene glycol) (PEG) through a two-step polycondensation method (transesterification and polycondensation). The copolymers are named PEGF-ØPEG-*M*nPEG, where *M*n is the molecular weight of PEG and ØPEG% is the theoretical weight percentage of PEG in copolymers. DMFD and EG with the molar ratio of 1:1.8 were placed into a three-necked round bottom flask. Then, Sb_2_(EG)_3_ (0.2%/mol DMFD), zinc acetate (0.15%/mol DMFD), heat stabilizer trimethyl phosphate (0.1%/mol DMFD), and antioxidants 1010 (0.1%/mol DMFD) were added. The flask was evacuated to 0.5 kPa and refilled with high purity nitrogen to remove the oxygen. The temperature of the reaction system was slowly raised to 90 °C under a nitrogen atmosphere to partially melt the DMFD. During the esterification reaction, the reactor was heated to 190 °C under a rate of 480 rpm/min stirring and holds for 3–4 h and 93% or more of methanol was distilled out. Finally, PEG was added before the vacuum was increased to 80–100 Pa and the temperature was raised to 235 °C. The polycondensation reaction lasted for 4 h. When an increasing effect occurred, the reaction ended. The target product PEGF copolyester was obtained.

Using the above method, this research varied the content of PEG (10, 20, 40, 60 wt %) and the number average molecular weight was 600, 2000, 6000, 10,000 g/mol to obtain a series of PEGF copolyesters.

### 2.3. Melt Spinning

#### 2.3.1. Purification of PEF and PEGF Copolyester

The PEF and PEGF-40%-2000 copolyesters were dissolved in a mixed solution of chloroform and trifluoroacetic acid in the ratio of 9:1. Methanol was used as a settling agent to precipitate a polyester product. The separated polyester was dried in a vacuum oven at 120 °C for 48 h to obtain purified PEF and PEGF-40%-2000 copolyester.

#### 2.3.2. Melt Spinning

We built a device ourselves, including screw extruder and winding machine. The ccrew extruder is to melt and extrude the polymer evenly. The winding machine continuously pulls the extruded polymer into fibers. The copolyester was spun by a micro single screw extruder and a winder. The specific experimental setup is shown in Figure 1. The single screw temperature is set as uniform and controlled at 200 °C, the winder speed was 20 m/min, and the temperature was raised to 200 °C. The PEGF-40%-2000 copolyester was added to the screw for melt spinning.

### 2.4. Measurements

#### 2.4.1. Intrinsic Viscosity

The PEGF copolyester was dried at 65 °C for 12 h using a vacuum oven and then cooled to room temperature in a desiccator. The intrinsic viscosities were tested at (25 ± 0.05) °C by an Ubbelohde viscometer (inside diameter of 0.7–0.8 mm). A total of 250 mg of PEGF samples were dissolved in a solvent mixture of phenol/tetrachloroethane (1/1, *w*/*w*). The intrinsic viscosity was obtained using Equations (1) and (2):(1)$$\eta_{sp}=\frac{t_{1}-t_{0}}{t_{0}}$$
(2)$$\;\;\eta =\frac{-1+\sqrt{1+1.4\eta_{sp}}}{0.7c}\;$$
where *η**_sp_* is the specific viscosity, *c* is the concentration (0.5 g/dL), and *t*_1_ and *t*_0_ are the flow times of the solution and pure solvent, respectively.

#### 2.4.2. Nuclear Magnetic Resonance (NMR) Analysis

Nuclear magnetic resonance (NMR) spectra were recorded on a 600 MHz machine manufactured by Bruker Switzerland (Fällanden, Switzerland) at 25 °C. The solvent was deuterated trifluoroacetic acid (TFA). The weight of the PEGF copolyester sample was 5–10 mg.

#### 2.4.3. Differential Scanning Calorimetry (DSC)

The thermal transition behavior was recorded by DSC under a N_2_ flow. A 5–10 mg aliquot of copolyester was weighed and put into a bowl. The sample was first heated from −40 to 250 °C with a 10 °C/min rate and maintained at 250 °C for 5 min to erase the thermal history; then, it was cooled to −40 °C at a rate of 10 °C/min rate. Finally, the second heating run was performed with heating at 10 °C/min to 250 °C.

#### 2.4.4. Thermogravimetric Analysis (TGA)

Thermal stability was investigated under a N_2_ atmosphere by a TGA instruction (USA TA/Q5000IR). Approximately 3 to 5 mg of the sample was continuously monitored from 40 to 600 °C at a rate of 10 °C/min. The variation in PEGF mass with temperature was recorded.

#### 2.4.5. Wide-Angle X-Ray Diffraction (WAXD)

Crystallization of PEGF copolyester was investigated by a Japanese max-2550VB+type 18kW target X-ray diffractometer. For the tests, a Cu target was used as an electrode light source, the voltage was 40 kV, the current was 150 mA, and the 2θ angle was 5 to 40°.

#### 2.4.6. Surface Contact Angle Test

The sample film was tested at 25 °C. For the tests, the amount of water dropped was 2 µL and the dropping rate was 1 µL/s. The contact angle value of the water droplets contacting the surface of the polyester film was recorded.

#### 2.4.7. Degradability Test

Degradation experiments were performed in a phosphate buffered solution (pH = 7.2) at 37 °C. The PEF and PEGF copolyester samples were prepared as films with the dimension of 1 cm × 3 cm × (0.1–0.3) mm. The samples were dried in a vacuum oven at 50 °C for 48 h. The dried samples were weighed separately and the mass was m_0_. The samples were placed in a phosphate buffer solution and a sample was taken every 10 days. Then, the quality and structure of the sample were characterized.

#### 2.4.8. Monofilament Strength

The PEF and PEGF-40%-2000 copolyester monofilaments were tested for strength using a monofilament strength instrument (WDW3020, Chongqing, China). Each sample was repeatedly measured 10 times, and the results were averaged.

## 3. Results and Discussion

### 3.1. Synthesis and Characterization of PEGF Copolyester 

In this study, bio-based 2,5-DMFD, PEG and EG were used as reaction materials and the polymerization scheme is shown in Figure 2. The reaction used different molecular weights of different PEGs (number average molecular weights of 600, 2000, 6000, 10,000 g/mol) and different PEG contents (10, 20, 40, 60 wt % relative to DMFD) of PEGF copolyester. The esterification rate during polymerization is shown in Table 2.

From the data in Table 1, when the esterification temperature is 190 °C, the esterification rate of each sample is above 90%. The intrinsic viscosity of PEGFs is shown in Table 1. The intrinsic viscosity of PEGF copolyester is between 0.62–0.99 dL/g, which is higher than the intrinsic viscosity of PEF (0.61 dL/g). When the content or molecular weight of PEG is increased, the intrinsic viscosity of PEGFs is increased, mainly because of the existence of flexible long-chain PEG, which increases the chain length and flexibility of PEGFs. These changes make part of the copolyester molecular chain tangle in the capillary, resulting in prolonged residence time of the solution in the capillary and indicating that the introduction of flexible PEG achieves a thickening effect on PEF.

The ^1^H NMR spectra of PEGFs are presented. The structure of PEGFs with different PEG contents when the number average molecular weight of PEG is 2000 g/mol is shown in Figure 3. The structure of PEGFs with different molecular weights when the PEG mass content is 40 wt % is shown in Figure 4. As seen in the figure, each copolyester contains a total of four different types of H atoms with different chemical environments, where *δ*_a_ = 7.32 ppm corresponds to the H atom on the furan ring, *δ*_b_ = 4.73 ppm corresponds to the H atom of the methylene group in the hard segment of the polyester, *δ*_c_ = 3.84 ppm corresponds to the H atom of the methylene group on the segment of PEG, and *δ*_d_ = 3.81 ppm corresponds to the H atom in the terminal hydroxyl group of each copolyester. Due to the different positions of –CH_2_– in the copolyester segment, the inductive effects of the surrounding atoms are different. The chemical shift of –CH_2_– at the chain end will shift. Therefore, there are small proton peaks near the peaks of b and c, where b2 is the proton peak of –CH_2_– in the PEF chain end, and c2 is the proton peak of –CH_2_– in the PEG segment. According to the NMR results, when the molecular weight of PEG is 2000 g/mol, the PEG content increases, and the area of the c peak gradually increases. The above information indicates that the copolyester obtained by the polymerization is the target product. 

The content of PEG in the copolyester was calculated according to Equations (3) and (4), and the results are shown in Table 1. According to the results of the calculation, the actual added PEG content is consistent with the nuclear magnetic calculation results, indicating that the actually added PEG is fully involved in the reaction.
(3)$$n({\mathrm{CH}}_{2}{\mathrm{CH}}_{2}O)\;=\frac{n_{DMFD}\;\times \;I_{c}}{2I_{a}}\;$$
(4)$$\;\frac{W\left(PEG\right)}{W\left(DMFD\right)}=\frac{n_{\left(-{\mathrm{CH}}_{2}{\mathrm{CH}}_{2}O\right)}\times 44}{36.83}\;$$

The chemical structure of the PEGFs was confirmed by FTIR. The infrared spectrum of the copolyester of different content of PEG when the number average molecular weight of PEG is 2000 g/mol is shown in Figure 5; the infrared spectrum of the copolyester of different PEG molecular weights when the content of PEG is 40 wt % is shown in Figure 6. The peaks at 2885 and 1115 cm^−1^ are attributed to the –CH_2_– and C–O–C vibration modes of PEG, respectively. In the copolyester, not only the characteristic peak of PEG, but also the characteristic peak of PEF appeared. The peak at 1720 cm^−1^ is attributed to the C=O stretching vibration. The stretching vibration peaks of C–O–C appear at 1260 cm^−1^. The stretching vibration peak of –CH_2_– is observed at 2962 cm^−1^. The C–H and C=C stretching vibration peaks on the furan are observed at 3120 and 1580 cm^−1^, respectively, and the bending vibration peaks of the furan ring appeared at 960, 828, and 760 cm^−1^. According to the ^1^H NMR and FTIR results, the positions of the characteristic peaks of the synthesized products are consistent with the positions of the characteristic peaks of each group in PEF and PEG. After the addition of PEG, the C–O–C peak of the product at 1115 cm^−1^ appeared to move to the low wavelength region. When the number of average molecular weight of PEG was 2000 g/mol, the peak width at that point increased with the increasing PEG content. When the molecular weight of PEG gradually increases, the peak widths change a little, indicating that when the molecular weight of PEG is 2000 g/mol, the obtained product is a PEGF copolyester. When the molecular weight of PEG is higher than 2000 g/mol, the resulting product may be a blend of PEGF copolyester and PEG, which remains to be further characterized.

### 3.2. Thermal Properties of PEGF Copolyesters 

The thermal stability of the PEGFs was characterized by TGA. The results are shown in Figure 7 and Figure 8. From Table 2, the thermal decomposition temperature of each copolyester is approximately 350 °C, the temperature at the fastest thermal decomposition rate is approximately 395 °C, and the remaining copolyester approximately 9% after the temperature is raised to 600 °C. Compared with that of PEF, the thermal stability of PEGFs is slightly decreased, but it is still much higher than the spinning temperature of conventional polyesters. Therefore, the addition of PEG does not affect its practical application in terms of thermal stability.

The crystallization behavior and melting performance of PEGFs were investigated by DSC. The crystallization and melting curves of PEGFs with different PEG contents (PEG *M*n = 2000 g/mol) are shown in Figure 9a,b. Compared with PEF, when the PEG content increases gradually, the melting peak area of PEGFs increases gradually. When the PEG content reaches 40 and 60 wt %, the copolyester exhibits an obvious crystallization peak during cooling, indicating that the increased PEG content can promote the crystallization of PEGFs and increase the crystallization rate. This is mainly because the copolyester uses a part of the shorter PEF segment as the crystal nucleus during the cooling process; thus the PEG segment with strong kinetic ability will be rapidly arranged on the crystal nucleus formed by the PEF, the crystal will gradually grow and the crystallization rate will be increased. As seen from Figure 9b, when the PEG content increases, the melting point of PEGFs gradually decreases. When the PEG content is 10 and 20 wt %, the crystallization peak appears before the melting of the PEGFs, indicating that when the PEG content in PEGFs is low, the crystallization rate of PEGFs is limited. In the cooling process, the PEGFs are not completely crystallized. During the heating process, when the temperature reaches the crystallization temperature again, the secondary crystals of PEGFs will appear, and imperfect crystallization peaks appear. When the PEG content is 40 and 60 wt %, the imperfect crystallization peak does not appear before the melting of the PEGFs, indicating that the crystallization rate of the PEGFs is faster with the increased PEG content, and that the crystallization is more perfect during the cooling process than before.

Figure 10a shows the cooling crystallization curves of PEGFs with PEG of different molecular weights (mass content: 40%). Compared with PEF, the addition of PEG promotes crystallization of the copolyester and increases its crystallization rate. When the molecular weight of PEG is 2000 g/mol, there is only one obvious crystallization peak during the cooling process of PEGFs. When the PEG molecular weight is 6000 and 10,000 g/mol, two crystallization peaks of PEGFs are observed during cooling, which is because the molecular weight of PEG of 2000 g/mol is equivalent to the molecular weight of PEF, so the two segments are well compatible. When the PEG molecular weight is 6000 and 10,000 g/mol, the PEG segment is long, and it is difficult for some PEG to begin the polycondensation reaction with PEF esterification during the polycondensation stage, resulting in some PEG coexisting with the copolyester in the form of a blend. During the cooling process, PEG present in a blended form is likely to undergo phase separation, so two crystallization peaks appear. Figure 10b shows the melting curve of PEGFs with PEG of different molecular weights (mass content: 40%). When the molecular weight of PEG increases gradually, the melting point of PEGFs increases first and then decreases, but always exceeds the melting point of PEF, indicating that the introduction of PEG segments destroys the regularity of PEF segments and decreases the melting point of PEGFs. When the molecular weight of PEG reaches 10,000 g/mol, substantial phase separation occurs in the PEGFs, which leads to a decrease in the melting point of the PEGFs.

### 3.3. Hydrophilicity of PEGF Copolyester

When PEF is applied to the fiber field, there are still problems such as poor hydrophilicity of the polyester. To improve the hydrophilicity of PEF, this research introduced a flexible PEG segment into the PEF segment to increase the amorphous area in the PEGF so that water molecules can easily penetrate into the structure of the PEGFs. The molecular weight and content of PEG all affect the hydrophilicity of PEGFs. In this research, the hydrophilicity of PEGFs with different PEG molecular weights (600, 2000, 6000, 10,000 g/mol) and different contents (10, 20, 40, 60 wt %) was characterized. The results are shown in Figure 11 and Figure 12.

Figure 11 shows the change in contact angle on the surface of PEGFs with different PEG contents (PEG *M*n = 2000 g/mol). As seen from Figure 11, when the molecular weight of PEG is the same, the surface contact angle of the PEGFs decreases from 80.1 to 63.3° with increasing the PEG content, but both are lower than the surface contact angle of the PEF polyester. This is mainly because after the introduction of PEG, the PEGFs form a block structure in which the hydrophilic and hydrophobic groups coexist, and the partial molecular chain of the PEG loosely arranges, so that surface water molecules can easily enter the surface of the PEGFs and increase the hydrophilicity of the PEGFs.

Figure 12 shows the change of contact angle on the surface of PEGFs with different PEG molecular weights (mass content: 40%). When the molecular weight of PEG increases, the surface contact angle of PEGFs decreases from 84.2 to 37.6°. Compared with the PEG content, the molecular weight of PEG has a greater influence on the hydrophilicity of PEGFs. The reason for this phenomenon is that when the molecular weight of PEG is 2000 g/mol, the length of the PEG segment is equivalent to the length of the segment of the PEF. In the polycondensation stage, the PEF is easily polycondensed with PEG to form a PEF-PEG copolyester. The hydrophilicity of PEGFs increases only because the PEG segment is loosely arranged, and there is a weak microphase separation between the soft and hard segments, which has less influence on the hydrophilicity of the copolyester. When the molecular weight of PEG reaches 10,000 g/mol, the molecular weight of PEF differs greatly from the molecular weight of PEG. During the polycondensation process, part of the PEF can react with PEG to form PEGFs, but some PEG still exists in the form of polymer blends. When the PEG segment increases, the molecular chain of the copolyester is loosely arranged, and there is a relatively substantial phase separation in the form of a blend of PEG and PEPFs, which greatly improves the hydrophilicity of the PEGFs. 

### 3.4. Degradability of PEGF Copolyester

Hydrolytic degradation of PEGFs was performed in a phosphate buffer solution (pH = 7.2) at 37 °C. Samples were taken every 10 days and dried in a vacuum oven at 50 °C for 48 h, and the mass of the sample was measured. Figure 13 shows the variation of the quality of PEF and its copolyester film with the degradation time. As seen from Figure 13, the quality change of PEF is not obvious during treatment in the phosphate buffer solution at pH = 7.2, and the mass loss is only approximately 0.5%, which fully indicates that under neutral conditions, PEF is not a degradable polymer. When PEG was introduced into PEF, the quality of PEGFs decreased with the prolongation of degradation time, which indicates that the introduction of PEG endowed the PEGFs with certain degradability, and the content and molecular weight of PEG had a great influence on the mass loss of PEGFs.

As seen from Figure 13a, when the molecular weight of PEG is 2000 g/mol, the weight loss rate of the PEGFs increases with increasing PEG content. When the PEG content is 60 wt %, the mass loss of the PEGFs is the highest, reaching about 15%. From the degradation results, the change in the degradability of PEGFs is consistent with the change in hydrophilicity of PEGFs. This is mainly because the introduction of PEG improves the hydrophilicity of PEGFs, and water molecules easily enter the interior of the molecule. The ester bond in the molecular chain is attacked by water molecules, a hydrolysis reaction occurs, and the molecular chain is broken.

As seen from Figure 13b, when the PEG content reaches 40 wt %, the PEGFs experience an obvious mass loss. With the increase in the molecular weight of PEG, the mass loss of PEGFs is more obvious and the degradation is more substantial. When the molecular weight of PEG is 10,000 g/mol, the mass loss of PEGFs is the highest, at 15.3%. When the molecular weight of PEG is 600 and 2000 g/mol, the change in quality of the two is basically the same as the degradation time increases. When the molecular weight of PEG is 6000 and 10,000 g/mol, the change in quality of both are the same. This is mainly because when the molecular weight of PEG is low, the length of the segment is equivalent to that of PEF. The compatibility between soft and hard segments is better, the attack probability of water molecules on ester bonds is not large, and the mass loss is relatively small. When the molecular weight of PEG reaches 6000 and 10,000 g/mol, the molecular chain lengths between the soft and hard segments are significantly different. These two are prone to serious phase separation, which increases the probability of attack by water molecules on ester bonds, and the quality loss is more substantial.

Table 3 shows the change in intrinsic viscosity after 100 days of degradation of PEGFs. From the table, after 100 days of degradation treatment, the intrinsic viscosity of PEF changes very little, only 0.1 dL/g, while the intrinsic viscosity of PEGF copolyester changes greatly. When the molecular weight of PEG is 2000 g/mol, the intrinsic viscosity of PEGFs increases with increasing PEG content. When the PEG content reaches 60 wt %, the intrinsic viscosity decreases by 0.66 dL/g. The conversion of PEGFs from high-molecular-weight polymers to low-molecular-weight polymers leads to the degradation of PEGFs. When the PEG content is 40 wt %, as the molecular weight of PEG increases, the intrinsic viscosity of PEGFs also decreases. When the molecular weight of PEG is 10,000 g/mol, the intrinsic viscosity of the PEGFs decreases by at most 0.45 dL/g. From the trend of intrinsic viscosity change, the introduction of PEG endows PEGFs with certain degradability, and its change trend is consistent with the trend of the change in quality of PEGFs.

Figure 14 shows the change in the pH value of the degradation solution after 100 days of degradation of PEGFs. With the prolongation of degradation time, the pH value of PEF and its copolyester degradation solution shows a decreasing trend, and the degradation solution gradually changes from neutral to weakly acidic, which indicates that the degradation mechanism of PEEFs is acidic hydrolysis. The mechanism is shown in Figure 15. As seen from Figure 14a, when the molecular weight of the PEG is 2000 g/mol and the PEG content is increased, the pH of the PEGF degradation solution is gradually increased. As seen from Figure 14b, with the increase of the PEG molecular weight, the pH value of the PEGF degradation solution is the same, which indicates that the content of PEG has a major influence on the pH value of the PEGF degradation solution. 

To further explain the degradation mechanism of PEGFs and the change in structure of PEGFs during hydrolysis, in this research, the most degraded samples (PEGF-60%-2000 and PEGF-40%-10000) were selected for the analysis of the surface topography of polyester film, nuclear magnetic resonance, and PEGF degradation experiments. Samples were selected after 0, 40, 70, and 100 days of PEGF degradation. 

As shown in Figure 16, the area was integrated to calculate the content of PEG in PEGFs. The results are shown in Table 4. As seen from Table 4, when the degradation time increases, the PEG content in PEGFs decreases gradually. Peaks a and b are integrated. The ratio of peak area do not change with the prolongation of degradation time, which indicates that during the degradation of PEGFs, the ester bond linking the PEG moiety is mainly hydrolyzed and broken, and the PEGFs are partially is degraded.

The degradation solution of the PEGF-60%-2000 copolyester was subjected to rotary evaporation at 70 °C to obtain a white flocculent material, which was subjected to nuclear magnetic resonance characterization. As seen from Figure 17, there are three proton peaks in the nuclear magnetic resonance curve, where *δ* = 4.79 ppm is the proton peak of deuterated water, and δ = 7.36 ppm is the proton peak in phosphate. By inspection, *δ* = 3.72 ppm is the proton peak of –CH_2_– in the PEG segment, which indicates that in the degradation of PEGFs, mainly occurs by PEG partial chain scission.

As seen in Table 4, at the beginning of degradation, the content of PEG changes slightly, but its mass loss is substantial. In the later stage of degradation, the content of PEG changes greatly, but its mass loss is small. In the beginning of degradation, although the copolyester has a certain hydrophilicity, the internal structure of the PEGFs is still very regular, and it is difficult for water molecules to interact with each ester bond. Some PEG-linked short-chain PEFs enter the degradation solution together, making the weight loss of the PEGFs more obvious. In the late stage of degradation, the molecular weight of PEGFs is low, and there are many pores inside the molecule. The ester bond linked to PEG is easily attacked by water molecules, which causes the ester bond at both ends of the PEG to break, allowing the PEG molecule to enter the degradation liquid, and the remaining short chain PEF is still present in the PEGFs. The mechanical properties of the PEGFs gradually deteriorate, showing the fragile nature of the material. The result is shown in Figure 19.

Figure 18 shows the nuclear magnetic resonance curve of the PEGF-40%-10000 degradation sample after treatment for different numbers of days. Table 5 shows changes in the PEG content. The PEG content in the PEGFs gradually decreases when the degradation time increases, but when the degradation time increases, the changes in the PEG content in each stage are almost the same. In the late stage of degradation, the change is slightly reduced. This phenomenon is completely different from that observed for the PEGF-60%-2000 copolyester. The reason for this phenomenon is that when the molecular weight of PEG reaches 10,000 g/mol, the segment of PEG is longer, and the PEF esterification chain length is much different from that of PEG. Thus, part of the PEG segment forms an ester with PEF, and part of the PEG segment forms a blend with PEGFs. With the prolongation of degradation time, the PEG segment in the form of blend is gradually degraded and dissolved in the degradation solution, and the PEG in the form of copolyester is gradually hydrolyzed and undergoes chain breakage during degradation. The mechanical properties of PEGFs were analyzed. When the degradation time is prolonged, the PEGF-40%-10000 copolyester is gradually transformed from a PEMF film with good toughness into a fragile film. 

The surface morphology of PEGFs was analyzed by SEM. Figure 19 shows SEM images of the PEF, PEGF-60%-2000, and PEGF-40%-10000 degradation films. With the prolongation of degradation time, there is almost no change in the surface of PEF, indicating that PEF cannot be degraded in a neutral buffer solution. For the PEGF-60%-2000 copolyester, during the pressing film formation, when the surface of the polyester film is naturally cooled from 240 °C to room temperature, the PEG segment rapidly crystallizes. When the degradation time is prolonged, the PEG segment on the surface of the polyester film is gradually reduced, and at the same time the surface has irregularities and a certain depth of pores. For the PEGF-40%-10000 copolyester, when the degradation continues for to 40 days, many pores appear on the surface of the PEGFs. As the degradation time is prolonged, the pores became larger and deeper, which indicates that the degradation of the PEGFs is more substantial.

### 3.5. Melt Spinning on PEF and PEGF-40%-2000

When the PEG content is 40 wt % and the molecular weight is 2000 g/mol, the obtained PEGFs have relatively excellent elasticity. In this research, a single-screw extruder and a small drafting device were used to spin the PEGF-40%-2000 to obtain the monofilament of the PEGFs. For comparison, the prepared PEF was spun and treated with a monofilament. The strength of the monofilaments is measured.

Figure 20 shows the PEF and PEGF-40%-2000 monofilaments. PEF shows large brittleness during spinning, so spinning is difficult to carry out. There are many broken filaments in the obtained monofilament. PEGF-40%-2000 has good elasticity during the spinning process, but the copolyester has a high viscosity and easily adheres to the mould. Thus, the fiber forming property is poor. The obtained monofilament has an uneven distribution and broken filaments. 

The mechanical properties of the monofilament were analyzed. The results are shown in Figure 21 and Table 6. As shown in Table 6, the PEF monofilament is difficult to draft, and its strength is low, only 1.15 cN/dtex. The monofilament shows high brittleness and poor mechanical strength. For the PEGF-40%-2000 copolyester, the monofilament can be stretched greatly at room temperature, the drafting ratio is four times, the fiber strength can reach 2.48 cN/dtex, and the elongation at break is 52.4%. Monofilaments have good elasticity and mechanical strength, but the spinnability of PEGF-40%-2000 still needs to be further improved.

## 4. Conclusions

In this work, PEGF copolyseters with different PEG molecular weights of 600 to 10,000 g/mol and different PEG contents were synthesized from DMFD, EG, and PEG. The effect of the PEG molecular weight and the PEG content on the structure and properties of the copolyester was analyzed. The intrinsic viscosity of the obtained copolyester was between 0.67 and 0.99 dL/g which is higher than the viscosity value of PEF. When the molecular weight of PEG is 2000 g/mol, the crystallization rate of copolyester increases with increasing PEG content. When the PEG content is 40 wt %, the crystallization rate increases most obviously, and the melting temperature and crystallization temperature decrease gradually. The hydrophilicity of PEGFs is improved (the surface contact angle is reduced from 91.9 to 63.3°), which endows PEGFs with a certain degradability, and the maximum mass loss can reach approximately 15%. When the PEG content is 40 wt %, the crystallization rate of PEGFs increases with increasing PEG molecular weight. The melting temperature and crystallization temperature first increase and then decrease, and the hydrophilicity of the PEGFs is improved (the surface contact angle is reduced from 91.9 to 37.9°), which gives the PEGFs a certain degree of degradability, and the mass loss can reach up to 15%. In this study, the fusion of PEF and PEGF-40%-2000 polyesters was analyzed for the first time. There are still many problems in the obtained fiber, such as uneven fiber distribution and the occurrence of broken wires and filaments. Therefore, further research on the polymerization conditions of PEF and its copolyester and the content and molecular weight of PEG are needed to produce bio-based fibers with certain spinnability and excellent mechanical properties.

## Figures and Tables

**Figure 1 polymers-11-02105-f001:**
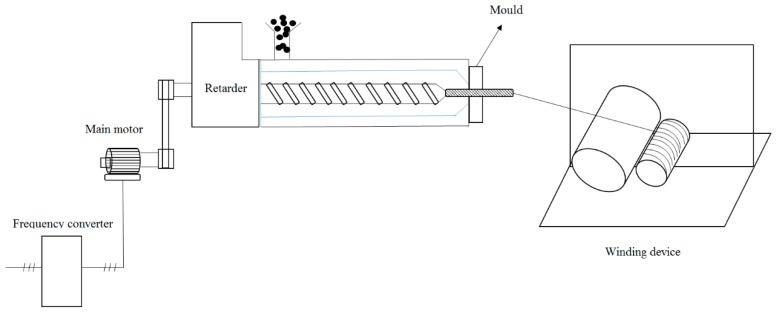
Melt spinning experimental device diagram.

**Figure 2 polymers-11-02105-f002:**
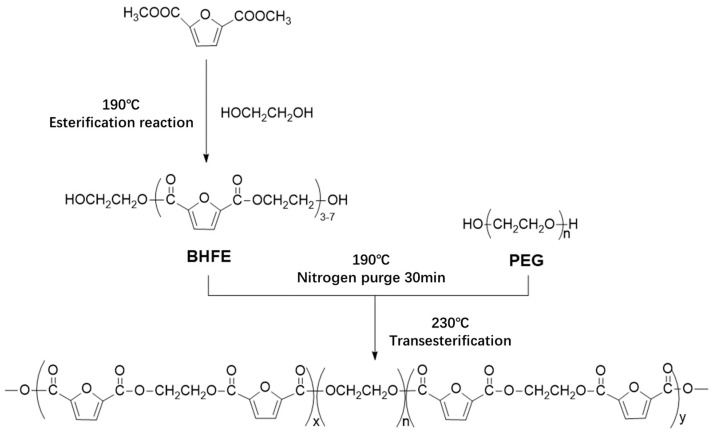
Polymerization scheme of PEF copolyester synthesized by transesterification.

**Figure 3 polymers-11-02105-f003:**
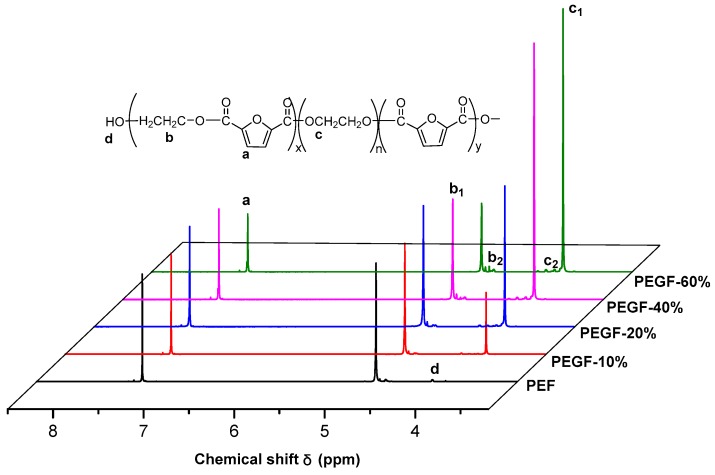
^1^H NMR spectra of PEGFs with different PEG contents (PEG *M*n = 2000 g/mol).

**Figure 4 polymers-11-02105-f004:**
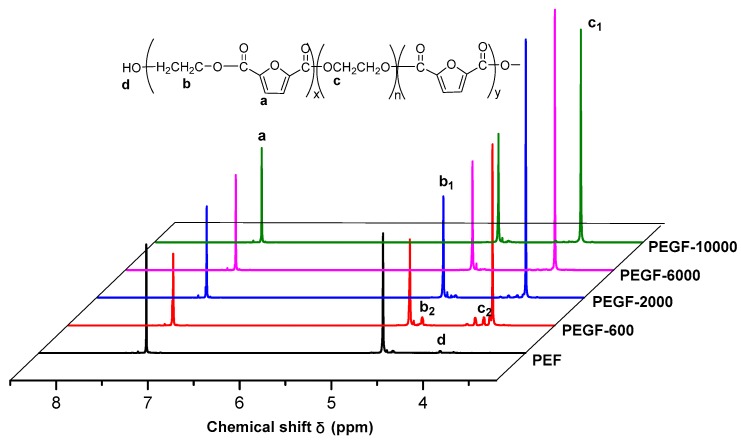
^1^H NMR spectra of PEGFs with PEG of different molecular weights (mass content: 40%).

**Figure 5 polymers-11-02105-f005:**
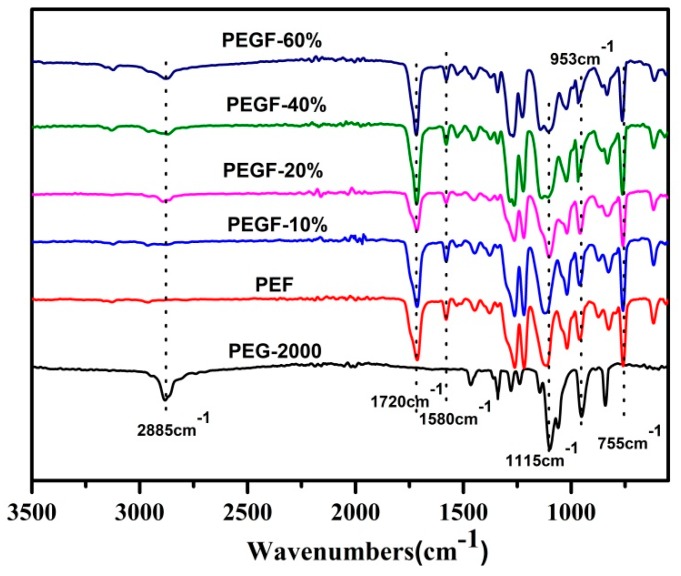
FTIR of PEGFs with different PEG contents (PEG *M*n = 2000 g/mol).

**Figure 6 polymers-11-02105-f006:**
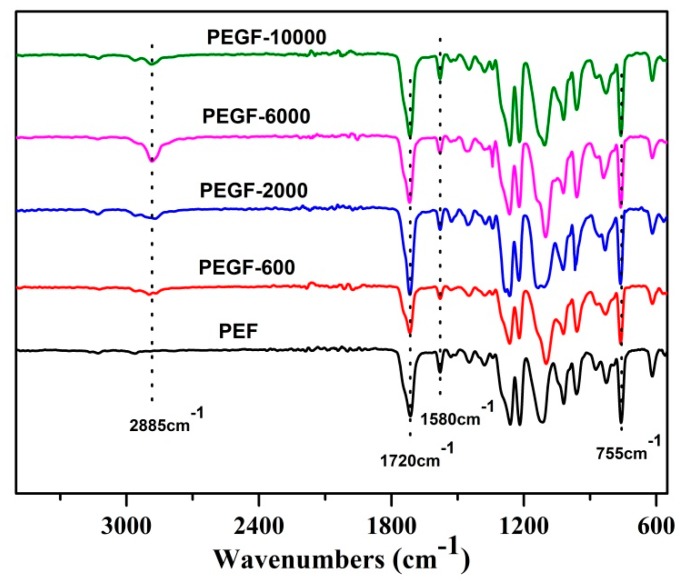
ATR FTIR of PEGFs with PEG of different molecular weights (mass content: 40%).

**Figure 7 polymers-11-02105-f007:**
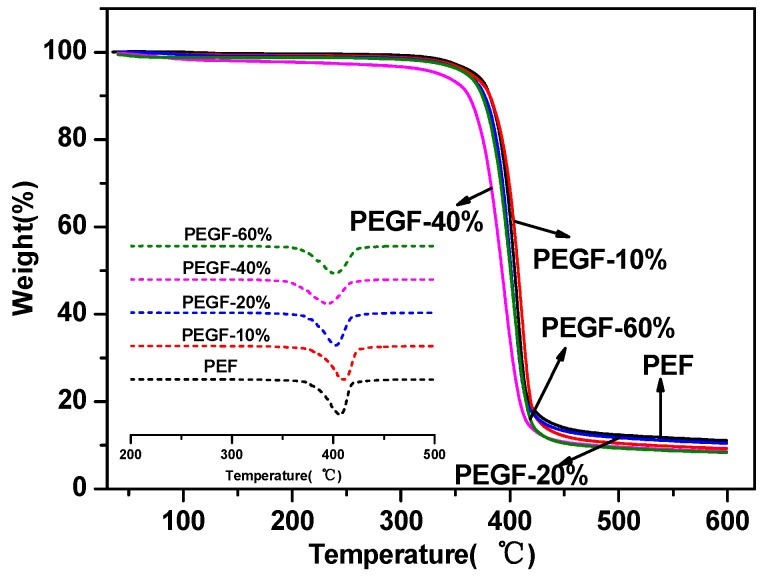
TGA of PEGFs with different PEG contents (PEG *M*n =2000 g/mol).

**Figure 8 polymers-11-02105-f008:**
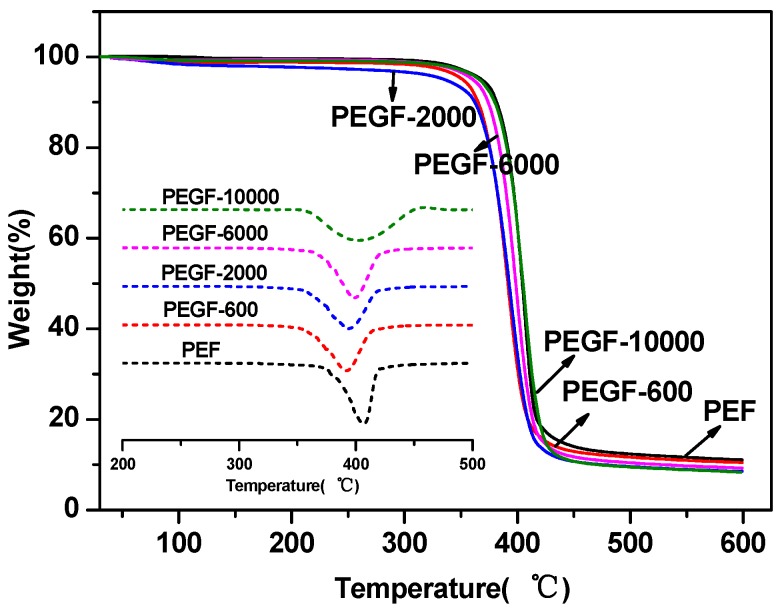
TGA of PEGFs with PEG of different molecular weights (mass content: 40%).

**Figure 9 polymers-11-02105-f009:**
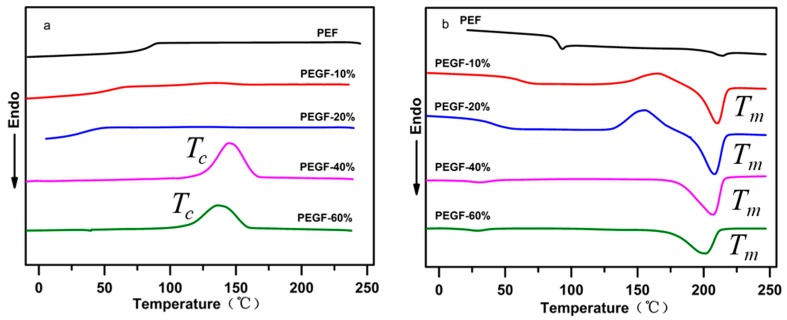
DSC of PEGFs with different PEG contents (PEG *M*n = 2000 g/mol). (**a**) Cooling crystallization curve of copolyester prepared for different molecular weight PEG. (**b**) Melting curve of copolyester prepared for different molecular weight PEG.

**Figure 10 polymers-11-02105-f010:**
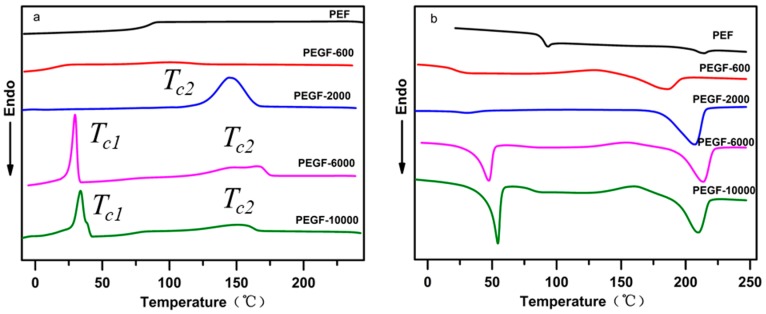
DSC of PEGFs with PEG of different molecular weights (mass content: 40%). (**a**) Cooling crystallization curve of copolyester prepared for different molecular weight of PEG. (**b**) Melting curve of copolyester prepared for different molecular weight of PEG.

**Figure 11 polymers-11-02105-f011:**
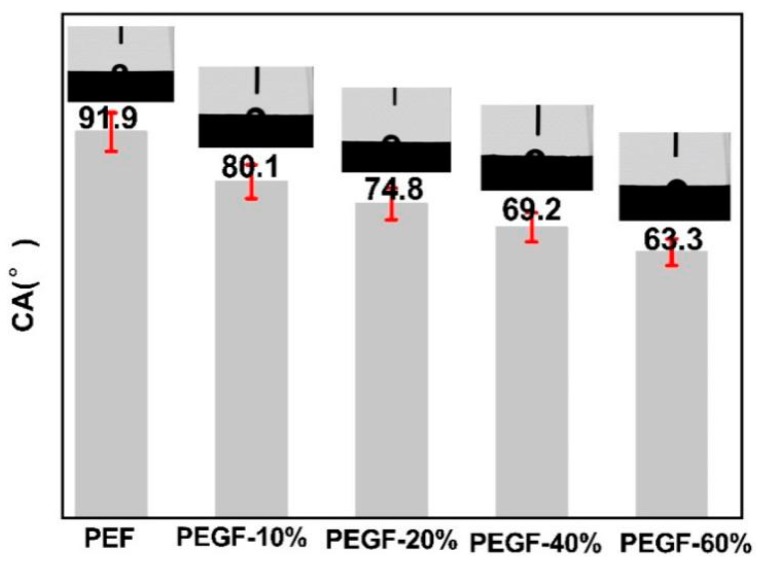
Contact angle of copolyester with different PEG contents (PEG *M*n = 2000 g/mol).

**Figure 12 polymers-11-02105-f012:**
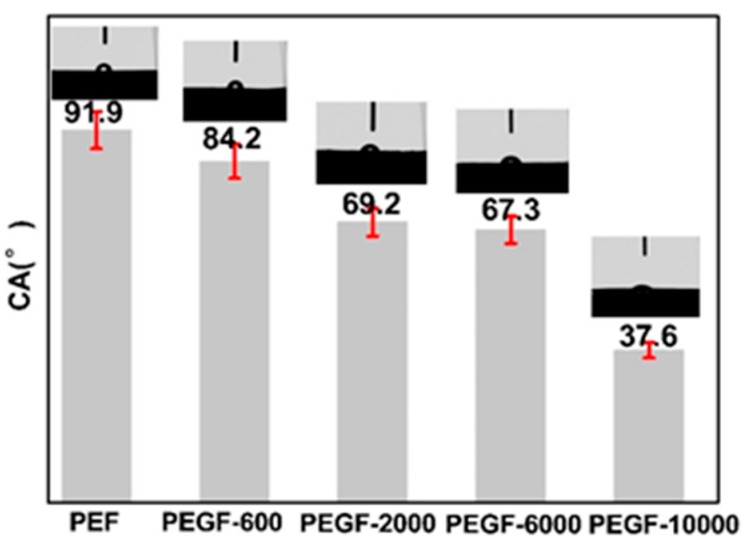
Contact angle of PEGFs with different PEG molecular weights (mass content: 40%).

**Figure 13 polymers-11-02105-f013:**
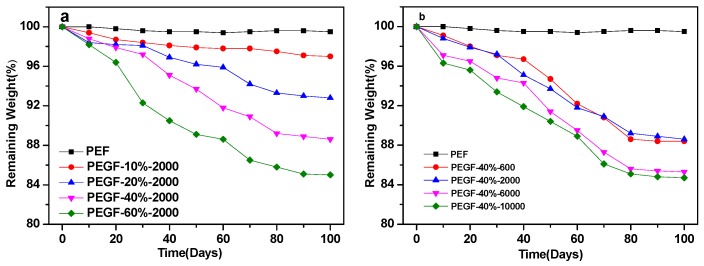
Mass change of PEF and PEGF copolyester (**a**: Different content of PEG (*M*n = 2000 g/mol) copolyester; **b**: Different molecular weights of PEG (mass content: 40%) copolyester).

**Figure 14 polymers-11-02105-f014:**
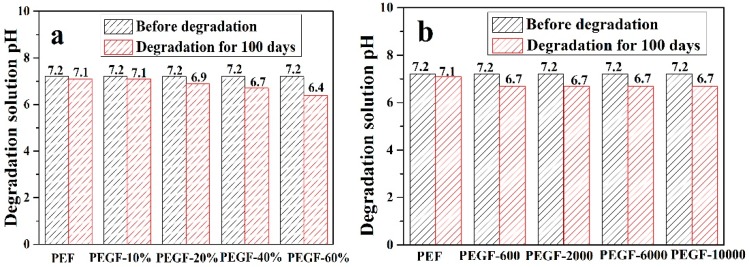
The change in the pH value of PEF and its copolyester degradation solution. (**a**: Different PEG contents (Mn = 2000 g/mol); **b**: Different PEG molecular weights (content: 40%))

**Figure 15 polymers-11-02105-f015:**
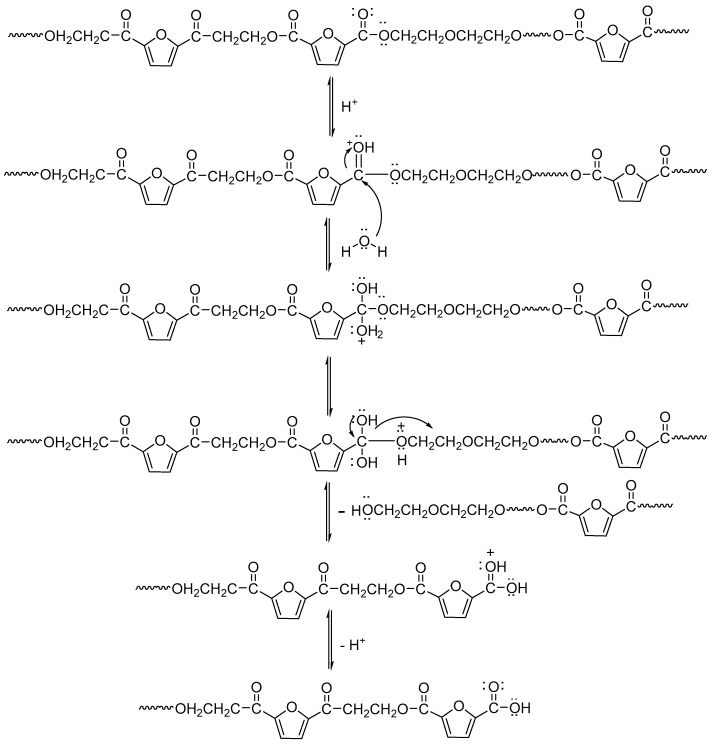
Copolyester hydrolysis mechanism diagram.

**Figure 16 polymers-11-02105-f016:**
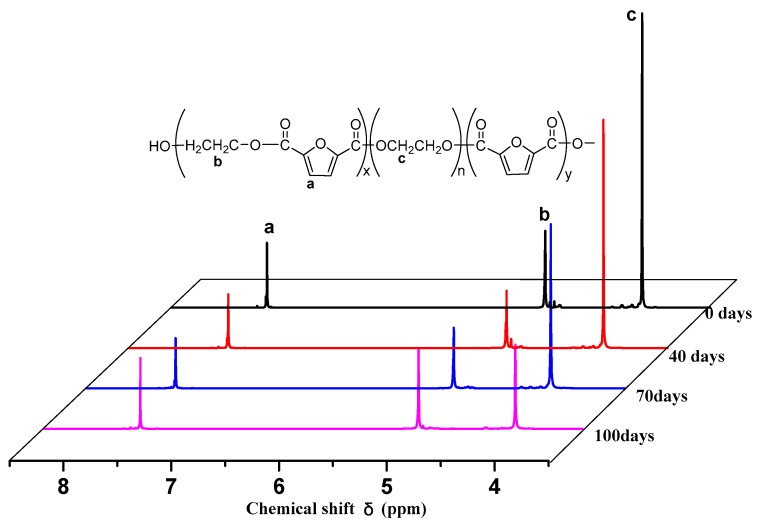
NMR of PEGF-60%-2000 degradation samples.

**Figure 17 polymers-11-02105-f017:**
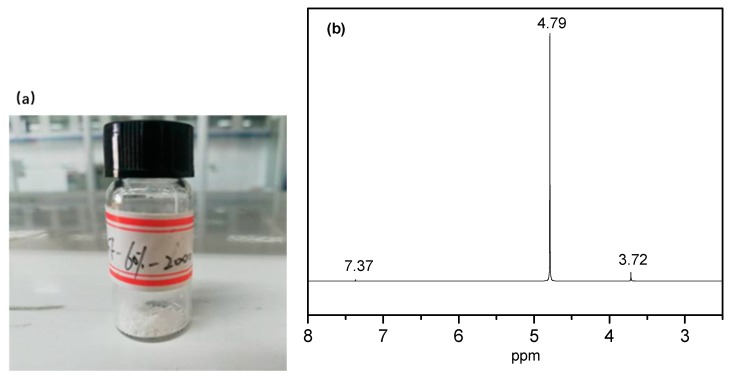
PEGF-60%-2000 degradation product (**a**) and nuclear magnetic resonance curve (**b**).

**Figure 18 polymers-11-02105-f018:**
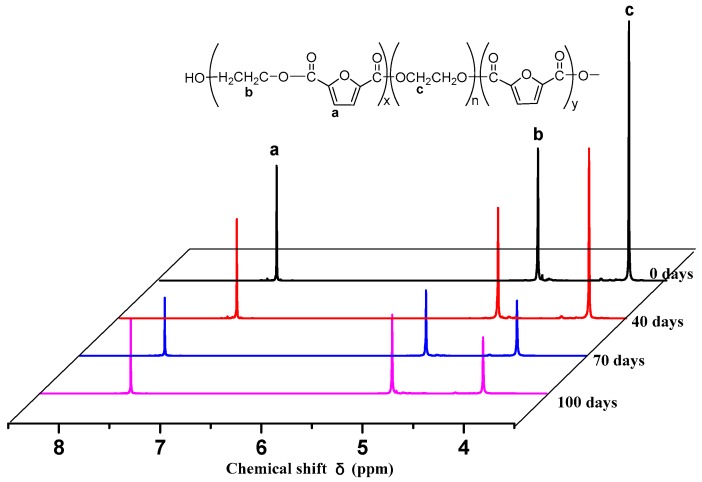
NMR of PEGF-40%-10000 degradation samples.

**Figure 19 polymers-11-02105-f019:**
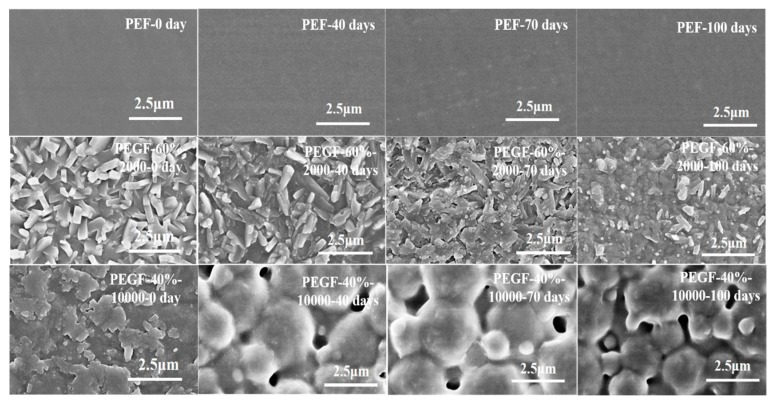
SEM images of PEF and its copolyester degradation samples.

**Figure 20 polymers-11-02105-f020:**
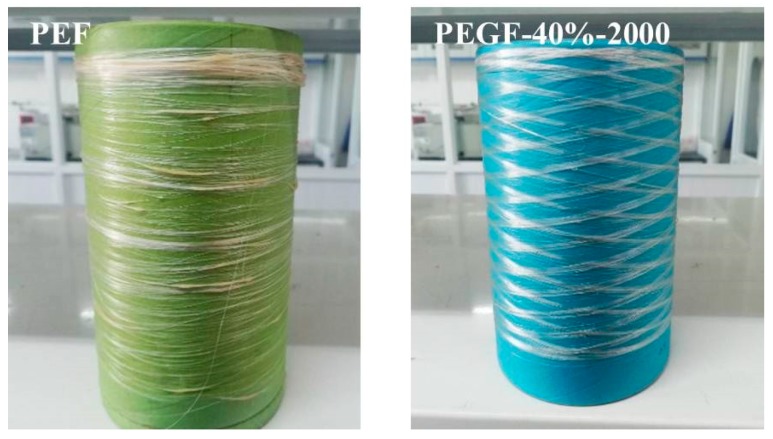
PEF and PEGF-40%-2000 monofilament.

**Figure 21 polymers-11-02105-f021:**
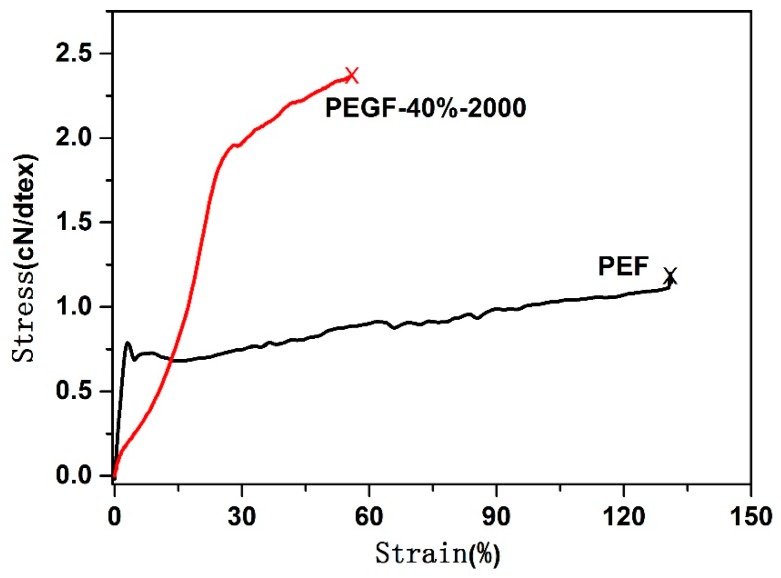
Tensile curve of the PEF and PEGF monofilaments.

**Table 1 polymers-11-02105-t001:** Physical properties of biobased PEGF copolyester.

Sample	NMR		Esterification Rate	Intrinsic Viscosity
	w(PEG)/w(DMFD) (Theory) (%)	w(PEG)/w(DMFD) (NMR) (%)	(%)	[*η*] (dL/g)
PEGF-10%-2000	10	9.7	91.5%	0.67
PEGF-20%-2000	20	19.5	93.8%	0.69
PEGF-40%-2000	40	39.8	91.7%	0.74
PEGF-60%-2000	60	58.3	90.9%	0.80
PEGF-40%-600	40	35.6	95.2%	0.62
PEGF-40%-6000	40	40.4	92.6%	0.76
PEGF-40%-10000	40	41.2	93.8%	0.99

Note: In PEGF-x%-y, x represents the content of PEG, and y represents the number average molecular weight of PEG.

**Table 2 polymers-11-02105-t002:** Thermodynamic parameters of PEGF copolyester.

Sample	DSC					TGA		
	*T*_g_(°C)	*T*_c_(°C)	Δ*H*_c_(J/g)	*T*_m_(°C)	Δ*H*_m_(J/g)	*T*_d_,_5%_(°C)	*T*_d_,_max_(°C)	*R*_600_(wt %)
PEF	90.84	--	--	214.04	1.97	368.23	407.19	11.07
PEGF-10%-2000	62.19	133.09	3.27	210.05	32.36	364.32	410.95	9.23
PEGF-20%-2000	40.25	--	--	207.88	27.96	360.14	402.08	10.44
PEGF-40%-2000	25.71	144.31	34.26	207.28	31.75	342.95	395.76	8.54
PEGF-60%-2000	23.91	136.17	26.28	201.44	24.42	355.01	400.73	8.34
PEGF-40%-600	20.93	100.93	4.24	185.15	25.40	351.09	392.13	10.48
PEGF-40%-6000	--	164.90	16.41	213.25	31.52	360.18	398.32	9.28
PEGF-40%-10000	--	151.53	8.43	209.25	28.57	365.35	405.84	8.36

*T*g (Glass transition temperature); *T*c (Crystallization temperature); Δ*H*c (Crystallization enthalpy change); Δ*H*m (Melting enthalpy change); *T*m (Melting temperature); *T*_d_,_5%_( Temperature at 5% mass loss); *T*_d_,_max_ (°C)(Temperature when decomposition rate is maximum); *R*600 (Residual amount).

**Table 3 polymers-11-02105-t003:** Changes in the intrinsic viscosity of PEF and its copolyesters over 100 days of degradation.

Sample	*η*_1_(dL/g)	*η*_2_(dL/g)	△*η* (dL/g)
PEF	0.61	0.60	0.01
PEGF-10%-2000	0.67	0.55	0.12
PEGF-20%-2000	0.69	0.41	0.28
PEGF-40%-2000	0.74	0.38	0.36
PEGF-60%-2000	0.80	0.14	0.66
PEGF-40%-600	0.62	0.35	0.27
PEGF-40%-6000	0.76	0.34	0.42
PEGF-40%-10000	0.99	0.54	0.45

Note: *η*_1_ is the intrinsic viscosity of the sample for 0 days of degradation; *η*_2_ is the intrinsic viscosity of the sample for 100 days of degradation; *△η = η*_1_ − *η*_2_.

**Table 4 polymers-11-02105-t004:** Change in PEG content with degradation time in PEGF-60%-2000.

Degradation Time (days)	w (PEG)/w(DMFD) Theoretical Value (%)	w(PEG)/w(DMFD) (NMR) (%)
0	60	58.3
40	60	55.3
70	60	48.9
100	60	26.8

**Table 5 polymers-11-02105-t005:** Change in PEG content in PEGF-40%-10000 with degradation time.

Degradation Time (days)	w(PEG)/w(DMFD) Theoretical Value	w(PEG)/w(DMFD)(NMR)
0	40%	41.2%
40	40%	33%
70	40%	25.2%
100	40%	19.5%

**Table 6 polymers-11-02105-t006:** Monofilament strength of PEF and PEGF-40%-2000.

Samples	Spinning Temperature (°C)	Drafting Ratio (Cold Drafting)	Linear Density (dtex)	Strength (cN/dtex)	Elongation at Break (%)
PEF	240	0	10.70	1.15	130.14
PEGF-40%-2000	200	4	9.30	2.48	52.40

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
