# Peer review of "Modification of Poly(Ethylene 2,5-Furandicarboxylate) with Poly(Ethylene glycol) for Biodegradable Copolyesters with Good Mechanical Properties and Spinnability"

_polymers, 2019, doi:10.3390/polym11122105_

Round 1

Reviewer 1 Report

line 78: purification.’ --> purification.

line 89 until 93, rephrase as some minor English formulation.

Line 94: this paper à suggest to change as this research

Section 2.3.2 please also explain parameter settings and type of equipment, as crucial in process

Line 114: viscometer(the --> viscometer (the 

Line 129 TG à TGA

Section 2.4.8 identify type of equipment

Line 163: Table 3-2, When --> Table 3-2, when 

Line 236: content:40%) -->content: 40%) 

Fig 3-12: CA(° ) --> CA (°) (as in text)

Line 365-369; rephrase as some errors

Section 3.5 explain processing and tst method including processing parameter settings and test settings.

Line 452 is --> conclusions

Within the whole work, mainly it is focusing on material development but finally spinning and testing is integrated. This last part is worthful to integrate, but some more details on process and methods of testing must be integrated. The process problems e.g. could be also reflected to rheology of newly polymers, but this is only tackled via IV, not via e.g. MFI or other rheology methods. Also conclusions must be expanded, reflecting to materials development and for sure processing.

Author Response

Dear Reviewer:

Thank you for your comments on our manuscript, entitled " Modification of poly(ethylene 2,5-furandicarboxylate) with poly(ethylene glycol) for biodegradable copolyesters with good mechanical properties and spinnability." (ID: 645082) These comments are valuable and useful for modifying and perfecting our papers, and have important guiding significance for our research. We have been carefully studied the comments and made corrections, hoping to get approval. The revised part is marked in red in the paper.

Point 1: Line 78: purification.’ --> purification.

Point 2: Line 94: this paper à suggest to change as this research

Point 3: Line 114: viscometer(the --> viscometer (the

Point 4: Line 129 TG à TGA

Point 5: Line 163: Table 3-2, When --> Table 3-2, when

Point 6: Line 236: content:40%) -->content: 40%)

Point 7: Fig 3-12: CA(° ) --> CA (°) (as in text)

Point 8: Line 452 is --> conclusions

 Response 1-8: Thank you very much for reading this manuscript and providing your valuable comments. Point 1-8 are to sort out your corrections about some of the format and general usage of this article. We are very sorry for our mistakes in writing. We have made correction according to the Reviewer’s comments. In order to avoid similar problems in the manuscript, we have extensively checked the format and writing specifications of the entire manuscript.

 Point 9: Line 89 until 93, rephrase as some minor English formulation.

Point 10: Line 365-369, rephrase as some errors

 Response 9-10: We are very sorry for the incorrect expression of some objective phenomena. This will cause some difficulties for readers to understand. We have made some revise in the corresponding position according to the Reviewer’s comments. In order to avoid similar problems in the manuscript, we reorganized the logical expression of the manuscript.

Point 11: Section 2.3.2 please also explain parameter settings and type of equipment, as crucial in process

Response 11: Thank you very much for your comments. We built a device ourselves, including screw extruder and winding machine.Screw extruder is to melt and extrude polymer evenl. The winding machine continuously pulls the extruded polymer into fibers. The copolyester was spun by a micro single screw extruder and a winder. The single screw temperature was set to 200℃, the winder speed was 20 m/min, and the temperature was raised to 200℃. The PEGF-40%-2000 copolyester was added to the screw for melt spinning.

 Point 12: Section 2.4.8 identify type of equipment

Response 12: The mechanical properties of the fiber were tested by fiber strength extensometer type of XQ-2 from Shanghai new fiber Instrument Co., Ltd. In the experiment, the clamping distance of the sample is 20 mm, the tensile rate is 20 mm / min, and the test temperature is room temperature. In the experiment, each group of samples was tested 10 times, and the results of strength, elongation at break and initial modulus were averaged.

 Point 13: Section 3.5 explain processing and tst method including processing parameter settings and test settings.

 Response 13: Thank you very much for your comments. We built a device ourselves, including screw extruder and winding machine.Screw extruder is to melt and extrude polymer evenl. The winding machine continuously pulls the extruded polymer into fibers. The copolyester was spun by a micro single screw extruder and a winder. The single screw temperature was set to 200℃, the winder speed was 20 m/min, and the temperature was raised to 200℃. The PEGF-40%-2000 copolyester was added to the screw for melt spinning.

Point 14: Within the whole work, mainly it is focusing on material development but finally spinning and testing is integrated. This last part is worthful to integrate, but some more details on process and methods of testing must be integrated. The process problems e.g. could be also reflected to rheology of newly polymers, but this is only tackled via IV, not via e.g. MFI or other rheology methods. Also conclusions must be expanded, reflecting to materials development and for sure processing.

Response 14: According to the Reviewer’s comments, we realized that our description of processing methods and processing parameters was insufficient. We have supplemented this part in the manuscript.

In the follow-up work, we will gradually optimize the process problems through the study of melt rheology. This manuscript focuses on the synthesis and degradation of new materials. Our application and outlook for this material has be added to the conclusions. In the future, the polymerization and spinning experiments will be optimeized.

If you have any better comments for our manuscript, please feel free to write to me again. We are very grateful for your comments on our paper.

Thank you and extend your sincere regards.

Kind regards,

Peng Ji

Reviewer 2 Report

One major suggestion I would like to provide is significantly improve the language of the manuscript. 

This manuscript is well planned and carried out, but content has to be revised for any subsequent consideration in polymers.

Based on the following reasons, I recommend minor revisions until the manuscript is improved for next round of consideration.

Please provide 13C-NMR spectra of the target compounds as well as the degradation compounds. Because the compounds are heavily carbon centric, 13C-NMR conclusive evidence for the hypothesis that were demonstrated by the 1H-NMR. Please provide stacked FTIR with enough clearance between each spectrum. Please mark heating and cooling cycles in the DSC Figures. Also mark representative thermal events (melting, crystallization). Please expand on the discussion on why low molecular weight (Mn) PEG addition causes low hydrophilicity, when low Mn addition is expected to improve hydrophilicity because of the abundant presence of hydrophilic OH groups. Although, exactly opposing trend was observed in this work. Can you plot all 1H-NMR without angular representation, and simply stack them?

General comment:

Please add a space between number and the units (°C, h, wt%, etc). Grammar and typographical errors are numerous. Only some of those listed below. Please improve the language thoroughly. Consider replacing reference 6 by the following: Ganesh Narayanan, Varadraj N. Vernekar, Emmanuel L. Kuyinu, Cato T. Laurencin. Poly (lactic acid)-based biomaterials for orthopaedic regenerative engineering, 107, 2016, 247-276. doi.org/10.1016/j.addr.2016.04.015 Also consider adding PCL, a widely used biodegradable suture material in the discussion. Also provide more concrete information like approx degradation times in-vivo

Specific Comments:

Revise the statement between the lines 35-38. Merge into a coherent statement. Line 41: replace “which severely restricted aliphatic polymers” by “severely restricting application of aliphatic polymers” Revise the statement between the lines 44-48. Either too short or too long statements. Line 57: replace “the introduce” by “the introduction of” Line 92: Define “climbing effect” line 99-100. Replace “chloroform:trifluoroacetic acid=9:1” by “chloroform and trifluoroacetic acid in the ratio of 9:1. Line 121: Bruker Line 125-126: at a heating rate of 10 ℃ /min Remove “It can record the variation of PEGFs mass with temperature” replace mass m by M Line 174: Define “4 different types of H”

12: throughout the manuscript, replace “In this paper” by “in this report”

Revise the statement between the lines “283-287”. Revise the statement between the lines “374-378”. Revise the statement between the lines “389-400”. Define “homoeothermy” Line 450. Completely revise conclusion section.

Author Response

Response to Reviewer 2 Comments

Dear Reviewer:

Thank you for your comments on our manuscript, entitled " Modification of poly(ethylene 2,5-furandicarboxylate) with poly(ethylene glycol) for biodegradable copolyesters with good mechanical properties and spinnability." (ID: 645082) These comments are valuable and useful for modifying and perfecting our papers, and have important guiding significance for our research. We have been carefully studied the comments and made corrections, hoping to get approval. The revised part is marked in red in the paper.

 Point 1: One major suggestion I would like to provide is significantly improve the language of the manuscript.

 Point 2: Please add a space between number and the units (°C, h, wt%, etc). Grammar and typographical errors are numerous. Only some of those listed below. Please improve the language thoroughly.

 Point 3: Revise the statement between the lines 35-38. Merge into a coherent statement. Line 41: replace “which severely restricted aliphatic polymers” by “severely restricting application of aliphatic polymers” Revise the statement between the lines 44-48. Either too short or too long statements. Line 57: replace “the introduce” by “the introduction of” Line 92: Define “climbing effect” line 99-100. Replace “chloroform:trifluoroacetic acid=9:1” by “chloroform and trifluoroacetic acid in the ratio of 9:1. Line 121: Bruker Line 125-126: at a heating rate of 10 ℃ /min Remove “It can record the variation of PEGFs mass with temperature” replace mass m by M Line 174: Define “4 different types of H”

Point 4: throughout the manuscript, replace “In this paper” by “in this report”

Response 1-4: Thank you very much for reading this manuscript and providing your valuable comments. We sort out your corrections about some of the format and general usage of this article. We are very sorry for our mistakes in writing. We have made correction according to the Reviewer’s comments. And mark the corresponding position in the manuscript. In order to avoid similar problems in the manuscript, we have extensively checked the format and writing specifications of the entire manuscript. Next, we will answer the professional questions in your comments one by one.

Point 5: Please provide 13C-NMR spectra of the target compounds as well as the degradation compounds. Because the compounds are heavily carbon centric, 13C-NMR conclusive evidence for the hypothesis that were demonstrated by the 1H-NMR.

Response 5: This is a very good suggestion, 13C-NMR conclusive evidence for the hypothesis that were demonstrated by the 1H-NMR. But due to the limitation of time and conditions, I will further supplement this experiment in the follow-up study.

Point 6: Please provide stacked FTIR with enough clearance between each spectrum.

Response 6: Thanks for the reviewer's comments, the drawing has been revised.

Point 7: Please mark heating and cooling cycles in the DSC Figures. Also mark representative thermal events (melting, crystallization).

Response 7:Thanks for the reviewer's very good suggestions. I have revised and supplemented them in the figure.

Point 8: Please expand on the discussion on why low molecular weight (Mn) PEG addition causes low hydrophilicity, when low Mn addition is expected to improve hydrophilicity because of the abundant presence of hydrophilic OH groups. Although, exactly opposing trend was observed in this work.

Response 8: Thanks for the reviewer's rigorous suggestions. We did this experiment repeatedly, and found that the hydrophilicity of the copolymers increased with the increase of the relative molecular weight of PEG. We also have reviewed the literature. With the increase of the molecular weight of polyethylene glycol, the polyethylene glycol segment and polyester segment in the copolymer will form their own phase structure, forming micro phase separation, which will lead to the improvement of hydrophilicity.

Point 9: Can you plot all 1H-NMR without angular representation, and simply stack them?

Response 9: Thanks for the reviewer's comments, the drawing has been revised.

Point 10: Please improve the language thoroughly. Consider replacing reference 6 by the following: Ganesh Narayanan, Varadraj N. Vernekar, Emmanuel L. Kuyinu, Cato T. Laurencin. Poly (lactic acid)-based biomaterials for orthopaedic regenerative engineering, 107, 2016, 247-276. doi.org/10.1016/j.addr.2016.04.015 Also consider adding PCL, a widely used biodegradable suture material in the discussion. Also provide more concrete information like approx degradation times in-vivo.

Response 10: Thank you very much for your comments. The language of the whole paper has been seriously revised. The reference “Poly (lactic acid)-based biomaterials for orthopaedic regenerative engineering” has been quoted in the paper. PCL is a very good and widely used biodegradable material. In the future, we will further compare the characteristics of different biodegradable polymers.

Point 11: Revise the statement between the lines “283-287”. Revise the statement between the lines “374-378”. Revise the statement between the lines “389-400”. Define “homoeothermy” Line 450. Completely revise conclusion section.

Response 11: Thanks for the reviewer's comments. The statement between the lines “283-287”, the lines “374-378” and the lines “389-400”. Define “homoeothermy” Line 450. Completely revise conclusion section. “homoeothermy” is to express “at room temperature”

If you have any better comments for our manuscript, please feel free to write to me again. We are very grateful for your comments on our paper.

Thank you and extend your sincere regards.

Kind regards,

Peng Ji

Reviewer 3 Report

The manuscript,  "Modification of Poly(ethylene 2,5-3 furandicarboxylate) with poly(ethylene glycol) for 4 Biodegradable copolyesters with good mechanical
5 and spinnability" by Ji et al. is well-writen and clearly characterized the resultant compounds. The impressive physical characterization and potential utility of these systems could be a good fit for this journal. I would go-ahead for publication with the article. 

Author Response

Response to Reviewer 3 Comments

Dear Reviewer:

Thank you for your comments on our manuscript, entitled " Modification of poly(ethylene 2,5-furandicarboxylate) with poly(ethylene glycol) for biodegradable copolyesters with good mechanical properties and spinnability." (ID: 645082) These comments are valuable and useful for modifying and perfecting our papers, and have important guiding significance for our research. We have been carefully studied the comments and made corrections, hoping to get approval. The revised part is marked in red in the paper.

If you have any better comments for our manuscript, please feel free to write to me again. We are very grateful for your comments on our paper.

Thank you and extend your sincere regards.

Kind regards,

Peng Ji

Round 2

Reviewer 1 Report

Thanks a lot for the improvement of the manuscript. Please note some minor remarks:

line 115 "evenl" must be evenly. 

As for temperature settings, it is advised to included "temperature is set uniform", instead this statement is not valid!

Author Response

Response to Reviewer 1 Comments (second round)

Dear Reviewer:

Thank you for your comments on our manuscript, entitled " Modification of poly(ethylene 2,5-furandicarboxylate) with poly(ethylene glycol) for biodegradable copolyesters with good mechanical properties and spinnability." (ID: 645082) These comments are valuable and useful for modifying and perfecting our papers, and have important guiding significance for our research. We have been carefully studied the comments and made corrections, hoping to get approval. The revised part is marked in red in the paper.

Point 1: line 115 "evenl" must be evenly.

Response 1: Thank you very much for your comments. It has been revised.

Point 2: As for temperature settings, it is advised to included "temperature is set uniform", instead this statement is not valid!

Response 2: Thank you very much for your professional opinion. For temperature setting, it is necessary to unify the temperature process parameters and then carry out relevant experiments. Based on these conditions, the results of the comparison will make sense.

If you have any better comments for our manuscript, please feel free to write to me again. We are very grateful for your comments on our paper.

Thank you and extend your sincere regards.

Kind regards,

Peng Ji